# CHANNEL-WISE MIXED-PRECISION QUANTIZATION FOR LARGE LANGUAGE MODELS

## ABSTRACT

Large Language Models (LLMs) have demonstrated remarkable success across a wide range of language tasks, but their deployment on edge devices remains challenging due to the substantial memory requirements imposed by their large parameter sizes. Weight-only quantization presents a promising solution to reduce the memory footprint of LLMs. However, existing approaches primarily focus on integer-bit quantization, limiting their adaptability to fractional-bit quantization tasks and preventing the full utilization of available storage space on devices. In this paper, we introduce Channel-Wise Mixed-Precision Quantization (CMPQ), a novel mixed-precision quantization method that allocates quantization precision in a channel-wise pattern based on activation distributions. By assigning different precision levels to different weight channels, CMPQ can adapt to any bit-width constraint. CMPQ employs a non-uniform quantization strategy and incorporates two outlier extraction techniques that collaboratively preserve the critical information, thereby minimizing the quantization loss. Experiments on different sizes of LLMs demonstrate that CMPQ not only enhances performance in integer-bit quantization tasks but also achieves significant performance gains with a modest increase in memory usage. CMPQ thus represents an adaptive and effective approach to LLM quantization, offering substantial benefits across diverse device capabilities.

## 1 INTRODUCTION

Large Language Models (LLMs), trained on massive text corpora and containing up to hundreds of billions of parameters, have demonstrated exceptional performance across a wide range of language tasks (Brown, 2020; Chowdhery et al., 2023; Du et al., 2022; Hoffmann et al., 2022; Thoppilan et al., 2022; Touvron et al., 2023a). However, deploying LLMs directly on devices for inference presents a significant challenge due to their enormous parameter sizes (Frantar & Alistarh, 2023; Xia et al., 2024; Xiao et al., 2024; Li et al., 2024b). For instance, GPT-3 (Brown, 2020), with 175 billion parameters, requires 350 GB of memory in FP16 precision, which far exceeds the capacity of the latest NVIDIA H100 GPU with 96 GB of memory, let alone the capabilities of edge devices.

Low-precision weight-only quantization (Park et al., 2024; Lee et al., 2024; Huang et al., 2024) has emerged as a promising solution to address this challenge by converting model weights from high bit-width representations (e.g., FP16) to lower bit-widths (e.g., 3-bit), significantly reducing the memory requirements for on-device LLM inference. In this work, we focus on Post-Training Quantization (PTQ) (Dettmers et al., 2022; Yuan et al., 2023; Shao et al., 2023; Liu et al., 2024), which quantizes the pre-trained models without the need for retraining – a process that is often costly and resource-intensive. Most existing PTQ methods concentrate on integer-bit quantization and employ a uniform low-bit-width representation across all layers (Lin et al., 2024; Chee et al., 2024; Frantar et al., 2022), as illustrated in Figure 1(a), yielding promising performance in low-bit quantization tasks. However, these methods are limited in their adaptability to devices with additional storage capacity. For example, they can only offer solutions constrained to integer-bit quantization (e.g., 2-bit) even if the device could support models with an average of, e.g., 2.2-bit precision, failing to utilize the extra storage space to further reduce quantization loss.

Mixed-precision quantization (Ma et al., 2023; Wang et al., 2019; Dong et al., 2019; Zhang et al., 2021) inherently supports fractional bit quantization by allowing model weights to be quantized at different precisions. However, few works have focused on mixed-precision quantization specifically

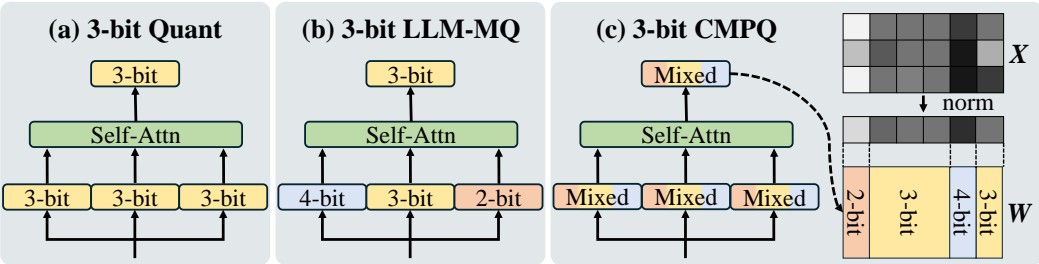

Figure 1: Illustration of different quantization approaches under a fixed bit-width constraint, such as 3 bits. (a) Standard quantization methods focus on algorithmic optimization to improve model performance, quantizing all layers uniformly to 3 bits. (b) LLM-MQ (Li et al., 2023) calculates layer-wise scores using first-order information and applies integer programming to assign lower bit-widths to less sensitive layers. (c) In contrast, our proposed CMPQ distributes the information loss evenly across layers by employing a channel-wise approach. This method assigns varying bit-widths within each layer based on activation distribution, ensuring that no single layer experiences significant information loss.

tailored for LLMs. LLM-MQ (Li et al., 2023) calculates the first-order information of layers at different bit-widths and uses integer programming to allocate layer-wise precision, as illustrated in Figure 1(b). Nevertheless, the gradient of a converged LLM is approximately zero, making it challenging for LLM-MQ to effectively differentiate the sensitivities of each layer. Furthermore, as demonstrated in Section 2, the additional quantization loss introduced by low-bit quantization can further degrade model performance.

To provide a mixed-precision method capable of adapting to any bit-width constraint without compromising the performance, we propose Channel-Wise Mixed-Precision Quantization (CMPQ). We first observe that different channels in the weight matrix have varying impacts on model performance, and that assigning higher precision to salient channels can enhance the performance of quantized LLMs. Inspired by this, CMPQ performs mixed-precision quantization on a channel-wise basis, as shown in Figure 1(c). Specifically, we compute the $L_2$-norm of the activation for each layer and allocate high (or low) precision to channels with large (or small) activation norms. For the quantization process, we adopt a non-uniform quantization approach for each channel, accounting for the non-uniform nature of weight distributions. To further improve performance, we design two outlier extraction methods that separately focus on preserving activation-based outliers and quantization-aware outliers. Our main contributions are summarized as follows.

- We propose CMPQ, a mixed-precision quantization method designed to adaptively quantize LLMs to any specified bit-width. CMPQ performs channel-wise quantization, allocating precision based on activation distributions to optimize the performance.
- CMPQ incorporates a non-uniform quantization strategy along with two outlier protection methods that collaboratively preserve critical weights in high precision, thereby reducing quantization loss.
- We conduct extensive experiments to empirically validate the effectiveness of CMPQ, highlighting two key advantages: (1) channel-wise quantization based on sensitivity significantly improves performance in integer-bit quantization tasks, and (2) a modest increase in storage overhead results in substantial performance gains.

## 2 RELATED WORKS

**Post-Training Quantization** Quantization is a model compression technique that modifies the vector or matrix representations of a pre-trained model to improve inference efficiency. It can be broadly categorized into two workflows: Quantization-Aware Training (QAT) (Liu et al., 2023; Xu et al., 2024; Li et al., 2024c; Dettmers et al., 2024) and Post-Training Quantization (PTQ) (Yao et al., 2022; Xiao et al., 2023; Huang et al., 2024; Hooper et al., 2024). QAT involves retraining the model while adjusting its parameters during quantization, which is resource-intensive and often impractical for LLMs. In contrast, PTQ focuses on quantizing pre-trained models without the need for retraining, making it a more feasible approach for resource-constrained scenarios. However, the complexity of

LLMs necessitates specialized approaches for effective quantization (Zhou et al., 2024). We focus on weight-only PTQ methods (Lee et al., 2024; Park et al., 2024; Dettmers et al., 2023; Tseng et al., 2024). One of the early advancements in this domain is GPTQ (Frantar et al., 2022), an improvement over OBQ (Frantar & Alistarh, 2022), which determines the optimal quantization order per row of the weight matrix based on reconstruction error relative to the Hessian matrix of unquantized weights. QuIP (Chee et al., 2024) further refines GPTQ by introducing an optimal adaptive method for a quadratic proxy objective. This method enhances quantization effectiveness by ensuring incoherence between the weight and Hessian matrices, achieved through random orthogonal matrix multiplication. AWQ (Lin et al., 2024) addresses the varying importance of weight channels for performance by employing a reparameterization technique, selecting coefficients via grid search to minimize reconstruction errors efficiently. SqueezeLLM (Kim et al., 2024) approaches quantization by storing outliers in a full-precision sparse matrix while applying non-uniform quantization to the remaining weights. Although these methods demonstrate promising performance in low-bit quantization tasks, they are limited in their adaptability to devices with additional storage capacity. Specifically, they cannot be easily extended to fractional-bit tasks, which prevents them from leveraging extra storage to further reduce quantization loss and improve model performance.

**Mixed-Precision Quantization** Quantizing models to low precision uniformly can lead to significant accuracy degradation (Habi et al., 2020; Qu et al., 2020; Hu et al., 2021). To address this, Mixed-Precision Quantization (MPQ) has emerged as a promising approach, aiming to reduce model size and computational costs while maintaining or even improving accuracy by assigning different bit widths to weights (Wang et al., 2019; Li et al., 2024a; Rakka et al., 2022). Existing methods primarily focus on relatively small DNNs, such as MobileNet (Wang et al., 2019; Howard, 2017) and ResNet (Wu et al., 2018; He et al., 2016), and argue that MPQ is necessary for different layers, as each layer exhibits varying levels of redundancy and performs differently on hardware (Wang et al., 2019). To determine the appropriate per-layer precision, these methods rely on techniques like Neural Architecture Search (NAS) (Wu et al., 2018), periodic function regularization (Naumov et al., 2018), and second-order sensitivity analysis (Dong et al., 2019; 2020; Yao et al., 2021). However, these approaches are difficult to generalize to the quantization of LLMs due to the extensive computational resources and large search space they require (Gholami et al., 2022). While protecting outliers in FP16 can be considered a form of mixed-precision, few studies have applied mixed-precision strategies to the majority of weights. LLM-MQ (Li et al., 2023) uses integer programming to allocate layer-wise precision based on first-order information. However, as discussed in Section 3.1.2, it struggles to capture the varying sensitivities of individual layers. To overcome the limitations of large search spaces and performance degradation, we propose a channel-wise mixed-precision strategy that protects salient channels within each layer, ensuring the preservation of critical information.

## 3 METHOD

In this section, we begin with preliminary studies to investigate the impact of channel-wise mixed-precision quantization compared to layer-wise mixed-precision quantization (Section 3.1.2). The empirical results align with our intuition: channel-wise mixed-precision quantization more effectively harnesses the potential of mixed-precision. Subsequently, in Section 3.2, we provide a detailed introduction to our proposed Channel-Wise Mixed-Precision Quantization method.

### 3.1 PRELIMINARY

#### 3.1.1 $N$-BIT QUANTIZATION

Quantization typically involves mapping a continuous set of values $\boldsymbol{W}$ from a higher bit-width (e.g., 16-bit floating point) to a discrete set of values $Q(\boldsymbol{W})$ at a lower bit-width (e.g., 4-bit integers). The most commonly used uniform quantization (Krishnamoorthi, 2018) can be expressed as:

$$Q(\boldsymbol{W}) = \left\lceil \frac{\boldsymbol{W}_{\text{FP16}} - \min(\boldsymbol{W}_{\text{FP16}})}{\Delta} \right\rfloor,$$

where $\Delta$ is the quantization step size, determined by the range of the original values $\boldsymbol{W}$ and the desired bit-width $N$. Specifically, $\Delta$ can be computed as:

$$\Delta = \frac{\max(\boldsymbol{W}_{\text{FP16}}) - \min(\boldsymbol{W}_{\text{FP16}})}{2^{N-1} - 1}.$$

However, LLMs typically exhibit non-uniform weight distributions (Kim et al., 2024; Dettmers et al., 2024). In such cases, the presence of large magnitude values can lead to inefficient quantization, where certain bins or bit combinations are underutilized, resulting in suboptimal representation with few or no values assigned to some bins. To address this issue and better preserve the original weight information, we adopt non-uniform quantization. For a given vector $\boldsymbol{x}$, we compute a $2^N$-bit quantized representation, denoted as $\boldsymbol{q} = \{q_1, \ldots, q_{2^N}\}$, and obtain its lower-precision approximation through the following procedure:

$$Q(\boldsymbol{x}) = (Q(x_i)), \text{ where } Q(x_i) = \min_{q_j \in \boldsymbol{q}} |x_i - q_j|.$$

### 3.1.2 PRELIMINARY STUDY OF MIXED-PRECISION QUANTIZATION

To explore the effect of mixed-precision quantization, we conduct preliminary experiments on two OPT models (Zhang et al., 2022) and compare the perplexity evaluation. We focus on 3-bit quantization, where precision is selected from {2, 3, 4}-bit.

In Li et al. (2023), each linear layer is associated with information loss scores under different precisions, and their approach models the average bit width as a constraint in an integer programming problem. We implement this strategy with the non-uniform quantization, and the performance is reported in Table 1 as **w/ IntProg**. Additionally, Lin et al. (2024) observe that retaining salient weights based on activation distributions in higher precision significantly enhances quantized performance. For each layer, we compute the channel-wise $L_2$-norm of activations and select the top (and bottom) $k\%$ of channels. These channels are quantized to 4-bit (high precision) or 2-bit (low precision) respectively. Results for $k = 1$ and $k = 10$ are presented in Table 1.

Table 1: Preliminary study of various mixed-precision 3-bit quantization methods. **w/ IntProg** denotes quantization using an integer programming solution for each layer. **w/ 10%2-bit, 10%4-bit** and **w/ 1%2-bit, 1%4-bit** indicate quantization where 10% (1%) of channels are 4-bit and 10% (1%) are 2-bit, based on the activation distribution. Best performances are in bold, with underlined text showing the second best.

| Method | OPT-2.7 | | OPT-6.7 | |
|---|---|---|---|---|
| | Wiki ($\downarrow$) | C4 ($\downarrow$) | Wiki ($\downarrow$) | C4 ($\downarrow$) |
| **3-bit** | 13.45 | 14.08 | 11.48 | 12.28 |
| **w/ IntProg** | 14.75 | 14.88 | 12.69 | 13.13 |
| **w/ 10%2-bit, 10%4-bit** | 13.53 | 14.27 | 11.61 | 12.42 |
| **w/ 1%2-bit, 1%4-bit** | **13.38** | **14.05** | **11.45** | **12.26** |

From Table 1, we observe that applying integer programming and assigning a fixed bit-width to each layer is insufficient for achieving effective mixed-precision quantization, and may even perform worse than using 3-bit quantization across all layers. This could be attributed to two factors: (i) the per-layer scores defined in Li et al. (2023) struggle to effectively distinguish the sensitivity of different layers, as the gradients in a converged LLM are nearly zero, and (ii) the additional information loss introduced by 2-bit quantization outweighs the compensatory gains from 4-bit quantization when compared to 3-bit quantization (Chee et al., 2024). On the other hand, quantizing each layer to different precisions based on activation distributions can yield better results than consistently using 3-bit across all layers. However, extending mixed-precision quantization to more channels could degrade performance, due to the same issue outlined in (ii).

## 3.2 CHANNEL-WISE MIXED-PRECISION QUANTIZATION

Our primary objective is to develop an algorithm that effectively utilizes mixed-precision quantization to adaptively compress LLMs under any given average bit constraint, including fractional bit-widths. Additionally, we still aim to achieve strong performance on integer-bit quantization tasks, such as 3-bit quantization, compared with existing works. To this end, we propose Channel-Wise Mixed-Precision Quantization (CMPQ). In this section, we first introduce the channel-wise non-uniform quantization method, followed by a detailed explanation of our outlier protection strategy.

### 3.2.1 Channel-Wise Non-Uniform Quantization

From the observations in Section 3.1.2, we can draw two key intuitions: ❶ channel-wise mixed-precision quantization can enhance model performance compared to layer-wise mixed-precision quantization, and ❷ when implementing channel-wise quantization, it is advisable to limit the number of 2-bit channels to minimize the information loss. Based on these, we propose channel-wise non-uniform quantization.

Research has shown that the weight distributions in LLM layers exhibit non-uniform patterns. Previous approaches have primarily focused on uniform quantization (Chee et al., 2024; Frantar et al., 2022), which divides the weight range into evenly spaced bins. However, this approach is suboptimal, as it fails to account for the non-uniform nature of the weight distributions, and struggles to improve end-to-end latency in memory-bound LLM inference (Kim et al., 2024). Following Kim et al. (2024), we adopt non-uniform quantization. Specifically, for each channel $W_{i,:}$ in the weight matrix $W \in \mathbb{R}^{d_{in} \times d_{out}}$, we apply a $K$-means clustering algorithm, where the value of $K$ is determined by the precision assigned to the channel (e.g., $K = 8$ for 3-bit quantization). After clustering, each weight in $W$ is represented by its nearest centroid from the set of $K$ centroids $\{q_1, \ldots, q_K\}$.

---

**Algorithm 1** Channel-wise Precision Allocation

---

**Input:** Activation norm vector $a \in \mathbb{R}^{d_{in}}$, average bit-width constraint $b \in [2, 4]$
**Output:** Channel precision allocation vector $c \in \mathbb{R}^{d_{in}}$
    Initialize $c = 3 \cdot \mathbf{1}$
    **if** $b > 3$ **then**
        Compute the $q$-th quantile $l$ of $a$, where $q = 1 - (b - 3)$
        Set $c[a > l] = 4$
    **else if** $b < 3$ **then**
        Compute the $q$-th quantile $s$ of $a$, where $q = 3 - b$
        Set $c[a < s] = 2$
    **else**
        Compute the 1st quantile $s$ and the 99th quantile $l$ of $a$
        Set $c[a < s] = 2$ and $c[a > l] = 4$
    **end if**

---

We draw inspiration from Lin et al. (2024) and our preliminary studies to determine the precision for each channel through a simple yet effective approach. We sample a calibration set to perform forward propagation on the LLMs, obtaining the activation matrix $X \in \mathbb{R}^{n \times d_{in}}$ for each layer weight matrix $W$. The per-channel $L_2$-norm of $X$ is then computed, yielding a 1-dimensional vector $a \in \mathbb{R}^{d_{in}}$. Based on this, we calculate the channel precision allocation vector $c$ as described in Algorithm 1, and apply non-uniform quantization to quantize each channel $W_{i,:}$ to $c_i$ bits. The key idea is that when the average bit-width $b$ exceeds 3, we focus on quantizing the most salient channels – determined by the activation distribution – into higher precision to enhance model performance. Conversely, when the average bit-width $b$ is below 3, we concentrate on quantizing less critical channels into lower precision to minimize quantization loss. For 3-bit quantization, we protect approximately 1% of the salient weight channels by assigning them 4-bit precision for improved performance, as motivated by the findings in Table 1.

It is worth noting that we intentionally avoid protecting additional channels with 4-bit precision and avoid incorporating equivalent channels in 2-bit quantization for compensation due to the following reasons: (i) Only a small fraction of weights are salient (Lin et al., 2024), and protecting them would significantly reduce quantization loss. (ii) The additional information loss incurred by 2-bit quantization outweighs the compensatory benefits gained from 4-bit quantization.

### 3.2.2 Outlier Protection

Another key challenge in low-bit LLM quantization is the protection of outlier values (Bondarenko et al., 2021; Wei et al., 2022; Dettmers et al., 2022). Previous studies have demonstrated that naively quantizing weights with a large dynamic range significantly degrades performance, particularly at low precisions (Kim et al., 2024). However, in some cases, retaining a small fraction (less than 1%) of

outlier weights in FP16 has been shown to reduce up to 75% of the total quantization error (Dettmers et al., 2023). This suggests that extracting outliers prior to quantization can mitigate their negative impact and minimize quantization loss. Consistent with prior works (Li et al., 2023; Kim et al., 2024), we retain 0.5% of outliers in high precision (16-bit), while applying quantization to the remaining weights. Our approach focuses on protecting two types of outliers: activation-based outliers and quantization-based outliers.

As illustrated in Figure 2, we observe that outliers exhibit a channel-wise pattern – if an outlier appears in a channel, it consistently occurs across all tokens. Table 1 empirically demonstrates that preserving salient weights based on the activation distribution helps mitigate quantization loss. Motivated by these findings, we introduce an activation-based outlier detection method, which identifies outliers $O_{\text{act}}$ from the weight matrix $W$. Specifically, we select channels corresponding to the top 0.45% largest values in the activation's L2-norm vector $a$, and preserve these channels in FP16 precision.

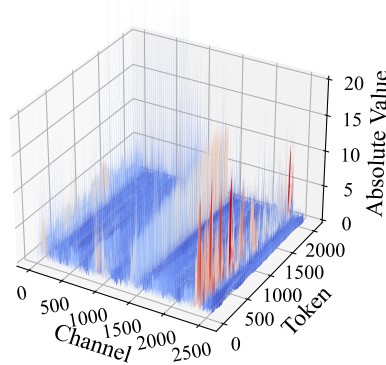

Figure 2: Magnitude of absolute activation values of the self_attn.out_proj layer in the third layer of OPT-2.7B.

In addition to selecting channel-wise outliers, we also investigate the protection of a small subset of quantization-sensitive outliers. Though we introduced our non-uniform quantization method, a small fraction of weights exhibit significantly larger magnitudes compared to the majority. These large weights can distort the clustering process by shifting centroids away from the bulk of the weight distribution, thereby negatively impacting the performance of the quantized LLM. A conventional approach to outlier protection involves the removal of weights based solely on their magnitude. However, instead of simply eliminating high-magnitude outliers, our objective is to identify and remove those that most adversely affect the quantization process.

To achieve this, we apply another $K$-means clustering step prior to quantization. Specifically, given $W' = W - O_{\text{act}}$ that represents the remaining weights after activation outlier removal, we use Algorithm 1 to determine the channel precisions $c$. We then apply channel-wise non-uniform quantization based on $c$ to obtain the quantized model $W'_q$. We identify the set of outliers $O_q$ in $W'$ corresponding to the top 0.05% of the largest values in $|W' - W'_q|$. This approach preserves these magnitude-based outliers in FP16 format, not only to mitigate their influence on model output but also to ensure that the centroids $\{q_1, \ldots, q_K\}$ better represent the majority of the weights, rather than being skewed by a small number of outliers. Two types of outliers $O = O_{\text{act}} + O_q$ are removed from the weight matrix $W$, and the remaining weights then undergo the quantization process to obtain $W_q$. $W_q + O$ is used for the final inference. Notably, the overhead associated with this decomposition is minimal, as the number of outlier values is relatively small, typically around 0.5% of the total values.

### 3.2.3 DISCUSSION

Our method only requires forward propagation and does not depend on backpropagation, which is necessary for many existing quantization techniques (Li et al., 2023; Kim et al., 2024). Consequently, the memory requirements for our proposed CMPQ during quantization are moderate; for instance, loading the OPT-6.7B model necessitates 12.4 GB of memory, whereas the backward pass for the same model requires 49.61 GB of memory. Additionally, CMPQ has minimal reliance on the calibration set, as it only measures the $L_2$-norm per channel, thereby mitigating the risk of overfitting. For a comparison with backpropagation-dependent methods, refer to Section 4.4, and for an analysis of CMPQ's robustness with respect to the calibration dataset, see Section 4.5.

## 4 EXPERIMENTS

### 4.1 EXPERIMENT SETUP

**LLM Models and Datasets.** We perform our experiments on two models from the OPT family (Zhang et al., 2022) (OPT-2.7B and OPT-6.7B) and two models from the LLaMA2 family (Touvron et al.,

2023b) (LLaMA2-7B and LLaMA2-13B). The evaluation of the quantized models is based on perplexity across two language generation tasks, WikiText-2 (Merity et al., 2016) and C4 (Raffel et al., 2020), as perplexity is a widely recognized metric for assessing the LLM performance. For calibration, we follow previous works (Chee et al., 2024; Frantar et al., 2022) and use a set of 128 randomly selected 2048-token segments from the C4 dataset, which contains generic text data from web crawls. This ensures consistency in comparison with baselines and avoids the use of task-specific data when quantizing other datasets. All experiments are implemented in PyTorch (Paszke et al., 2019) and executed on two A6000 GPUs, with performance monitoring handled by the Torch CUDA profiler. we extend our evaluation in Appendix A.3 to include quantization results for other OPT models, scaling up to 30B parameters, as well as a newer model, LLaMA3-8B (AI@Meta, 2024).

**Baselines.** We evaluate the proposed CMPQ against several post-training quantization methods that do not rely on backpropagation, including Round-to-Nearest (RTN), GPTQ (Frantar et al., 2022), AWQ (Lin et al., 2024), and QuIP (Chee et al., 2024). Since these methods are specifically designed for integer bit-width quantization, we restrict the comparison to {2, 3, 4}-bit settings. Additionally, we compare CMPQ with a mixed-precision quantization method tailored for LLMs, LLM-MQ (Li et al., 2023), focusing on performance in fractional bit-width quantization. In Section 4.4, we extend the comparison to also include SqueezeLLM (Kim et al., 2024), a state-of-the-art gradient-based method, and discuss the trade-offs between memory cost and quantization performance.

## 4.2 MAIN RESULTS

Table 2 presents the main results comparing CMPQ with post-training quantization baselines. Overall, our proposed CMPQ consistently outperforms all baselines, particularly in the 2-bit quantization tasks. Notably, while QuIP is specifically designed for low-bit quantization, it performs poorly on the WikiText-2 task using 2-bit quantized small LLM models (OPT-2.7B). In contrast, CMPQ achieves significantly better results on this task, despite using the same calibration dataset from C4. This highlights that CMPQ is less sensitive to the choice of the calibration set, as it relies solely on measuring the activation per-channel $L_2$-norm, which generalizes more effectively across different dataset distributions. Furthermore, although LLM-MQ is designed for mixed-precision quantization, it struggles to achieve competitive performance in 3-bit quantization compared to other baselines. This limitation arises from its precision allocation strategy, which is based exclusively on first-order information that is hard to distinguish layer sensitivities of a converged LLM.

## 4.3 NON-INTEGER QUANTIZATION

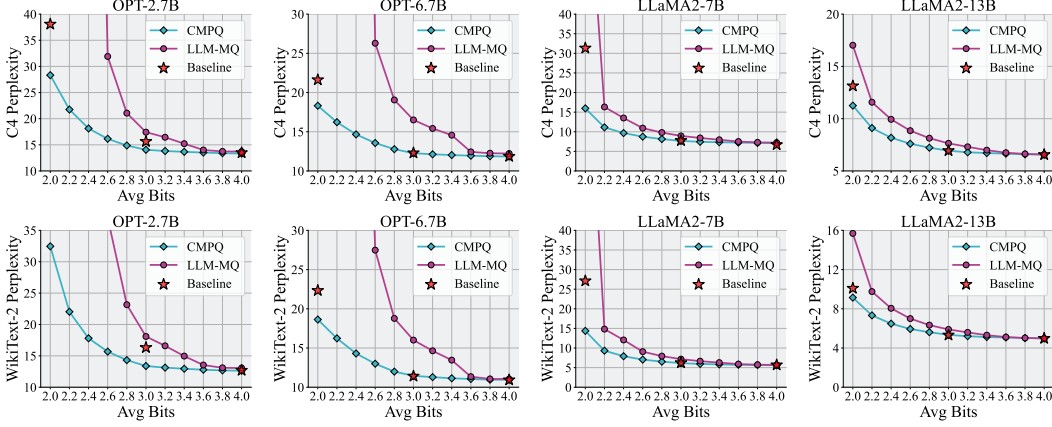

Figure 3: Comparison of C4 perplexity between CMPQ and LLM-MQ for fractional bit-width quantization. The red star indicates the best performance achieved by integer-based baselines at {2, 3, and 4} bits.

In Figure 3, we compare the perplexity of LLMs quantized by CMPQ and LLM-MQ under fractional bit-width constraints. First, it is evident that CMPQ consistently outperforms LLM-MQ across various bit-widths and models. LLM-MQ quantizes entire layers to a fixed precision, which can result in significant information loss within a single layer, negatively impacting the overall model

Table 2: Perplexity (↓) comparison of OPT and LLaMA2 models quantized to {2, 3, 4}-bit precision using various quantization methods on the C4 and WikiText-2 datasets. Bold font indicates the best performance across all methods, while underlined results denote the second-best.

| Method | Avg. Bit | OPT-2.7B | | OPT-6.7B | | LLaMA2-7B | | LLaMA2-13B | |
| --- | --- | --- | --- | --- | --- | --- | --- | --- | --- |
| | | Wiki | C4 | Wiki | C4 | Wiki | C4 | Wiki | C4 |
| **FP16** | 16 | 12.47 | 13.17 | 10.86 | 11.74 | 5.47 | 6.97 | 4.88 | 6.47 |
| **RTN** | 2 | >100 | >100 | >100 | >100 | >100 | >100 | >100 | >100 |
| **GPTQ** | 2 | >100 | >100 | >100 | >100 | 36.77 | 35.7 | 13.67 | 16.45 |
| **AWQ** | 2 | – | – | – | – | >100 | >100 | >100 | >100 |
| **QuIP** | 2 | >100 | 38.07 | 22.33 | 21.62 | 27.12 | 31.33 | 10.09 | 13.13 |
| **LLM-MQ** | 2 | >100 | >100 | >100 | >100 | 96.61 | 85.16 | 15.69 | 17.02 |
| **CMPQ** | 2 | **32.46** | **28.32** | **18.63** | **18.31** | **14.37** | **15.97** | **9.14** | **11.25** |
| **RTN** | 3 | >100 | >100 | >100 | >100 | >100 | >100 | 10.68 | 12.50 |
| **GPTQ** | 3 | 17.09 | 18.14 | 14.87 | 17.13 | 6.25 | 7.97 | 6.17 | 7.06 |
| **AWQ** | 3 | 16.32 | 16.28 | **11.41** | 12.30 | 6.24 | 7.84 | **5.32** | 6.94 |
| **QuIP** | 3 | 17.44 | 15.63 | 11.51 | 13.30 | 6.80 | 7.75 | 5.65 | 7.25 |
| **LLM-MQ** | 3 | 18.09 | 17.42 | 16.01 | 16.51 | 7.16 | 8.94 | 5.89 | 7.64 |
| **CMPQ** | 3 | **13.38** | **14.05** | 11.45 | **12.26** | **6.14** | **7.66** | 5.34 | **6.93** |
| **RTN** | 4 | 16.69 | 18.75 | 12.15 | 14.40 | 6.12 | 7.72 | 5.20 | 6.83 |
| **GPTQ** | 4 | 12.93 | 14.99 | 11.49 | 13.16 | 5.72 | 7.23 | 5.08 | 6.74 |
| **AWQ** | 4 | 12.73 | 13.48 | **10.93** | 11.86 | 5.72 | 7.13 | 4.98 | 6.56 |
| **QuIP** | 4 | 12.69 | 14.55 | 10.98 | 12.86 | 5.72 | **6.69** | 5.29 | 6.83 |
| **LLM-MQ** | 4 | 13.06 | 13.70 | 11.04 | 12.22 | 5.68 | 7.22 | 4.98 | 6.58 |
| **CMPQ** | 4 | **12.63** | **13.33** | 10.95 | **11.83** | **5.61** | 7.10 | **4.98** | **6.55** |

performance. In contrast, CMPQ allocates mixed-precision in a channel-wise manner, distributing the information loss more evenly across layers and avoiding a substantial loss in any single layer. Another key observation, visible when examining the transition from 2-bit to 2.2-bit quantization, highlights the advantage of mixed-precision. Introducing just a 10% increase in storage overhead at lower bit-widths can lead to significant performance gains. For instance, in the case of LLaMA2-7B, CMPQ yields a 30% improvement in perplexity (from 15.97 to 11.11), while LLM-MQ also shows a dramatic improvement, moving from poor performance (85.16) to performance that even surpasses the baseline (16.32). This demonstrates that mixed-precision quantization can trade a small increase in storage overhead for a substantial boost in performance – something that is not achievable with integer-only bit quantization methods.

As the average bit-width increases, the performance of both methods converges and approaches that of the baseline at 4-bit quantization. This indicates that quantization techniques face greater challenges at lower bit-widths. The strong performance of CMPQ at 2-bit and 3-bit quantization demonstrates its effectiveness, particularly in scenarios where lower bit-widths are required.

## 4.4 COMPARISON WITH SQUEEZELLM

While we primarily compare with baselines that do not rely on backpropagation, we acknowledge that gradient information, at the cost of extra resources, can indeed enhance the performance of quantized LLMs. In Table 3, we compare our method with the state-of-the-art baseline, SqueezeLLM (Kim et al., 2024), which also employs non-uniform quantization but uses gradient information to weight the clustering process, safeguarding more sensitive weights. Additionally, we report the memory requirements for loading LLMs and performing backpropagation in FP16 precision[1].

As shown in Table 3, SqueezeLLM outperforms CMPQ across various models at different quantization levels. However, this improvement comes at a significant cost: SqueezeLLM requires four times the memory for the quantization process, making it impractical for larger models, especially under

---

[1]The memory requirements were calculated using the Hugging Face platform (https://huggingface.co/spaces/hf-accelerate/model-memory-usage).

Table 3: Comparison of WikiText-2 perplexity between SqueezeLLM (abbreviated as SqzLLM) and CMPQ across various average bit-widths and models. The Memory column includes the total model size and memory requirements for backpropagation with a batch size of 1. For CMPQ, we also report performance at 2.2-bit and 3.2-bit to facilitate trade-off discussions. Bold font indicates the best performance across all methods, while underlined results denote the second-best.

| Models | Memory (GB) | SqzLLM 2 | CMPQ 2/2.2 | SqzLLM 3 | CMPQ 3/3.2 | SqzLLM 4 | CMPQ 4 |
|---|---|---|---|---|---|---|---|
| **OPT-2.7B** | 4.94/19.76 | – | 32.46/**22.03** | 13.43 | 13.38/**13.12** | **12.60** | 12.63 |
| **OPT-6.7B** | 12.40/49.61 | – | 18.63/**16.23** | 11.31 | 11.45/**11.29** | **10.92** | 10.95 |
| **LLaMA2-7B** | 12.37/49.48 | 10.79 | 14.37/**9.32** | 5.96 | 6.14/**5.95** | **5.57** | 5.61 |
| **LLaMA2-13B** | 24.02/96.07 | 7.91 | 9.14/**7.33** | 5.23 | 5.34/**5.21** | **4.96** | 4.98 |

resource constraints. In contrast, CMPQ offers a more efficient solution when memory is limited, requiring only 1/4 of the computational resources. Moreover, if an additional 10% of storage space is available, CMPQ can achieve better performance than SqueezeLLM, particularly in low-memory environments. At higher precision, such as 4-bit quantization, the tradeoff between computational resource requirements and model storage is less pronounced. The maximum performance gain of SqueezeLLM over CMPQ is marginal – only $(5.61 - 5.57)/5.61 = 0.71\%$. This minimal improvement renders the substantial additional resource demands of SqueezeLLM unnecessary.

Overall, while CMPQ may not surpass SqueezeLLM in the integer-only quantization tasks, the method offers significant advantages by eliminating the 300% increase in computational overhead associated with SqueezeLLM's gradient-based approach, with only a 10% increase in storage. Furthermore, CMPQ is more versatile, as it applies to non-integer bit quantization tasks, making it a more practical option for a wider range of scenarios.

### 4.5 DATA EFFICIENCY AND GENERALIZATION

**Data Efficiency for the Calibration Set.** In Table 4, we present a data efficiency analysis based on the number of data samples in the calibration datasets and compare the perplexity of the LLaMA2-7B model under 3-bit quantization. Although a calibration set of 128 data samples is used consistently throughout the paper, our method typically achieves the desired quantization performance with as few as single-digit sample sizes. This efficiency stems from the fact that we do not rely on regression or backpropagation; instead, we only measure the activation norm from the calibration set, making the process highly data-efficient. In contrast, both GPTQ and AWQ require more than 50 data points for calibration, as reported in (Kim et al., 2024).

Table 4: Data efficiency analysis of calibration datasets: Comparing CMPQ and SqueezeLLM for perplexity on C4 and WikiText-2 with 3-bit quantization of the LLaMA2-7B across varying calibration set sizes.

| Methods | Tasks | Nmuber of calibration samples | | | | | |
|---|---|---|---|---|---|---|---|
| | | 1 | 2 | 5 | 10 | 20 | 100 |
| **SqzLLM** | Wiki | 6.41 | 6.22 | 6.20 | 6.16.7 | 6.16 | 6.18 |
| | C4 | 7.89 | 7.81 | 7.73 | 7.72 | 7.72 | 7.72 |
| **CMPQ** | Wiki | 6.18 | 6.14 | 6.16 | 6.14 | 6.14 | 6.14 |
| | C4 | 7.71 | 7.65 | 7.66 | 7.65 | 7.65 | 7.66 |

**Robustness to the Calibration Set.** We evaluate the robustness of CMPQ by analyzing its performance using different calibration sets. Specifically, we compare CMPQ with QuIP in quantizing two OPT models into 2-bit representations. The results, presented in Table 5, demonstrate that CMPQ consistently outperforms QuIP in low-bit quantization across different calibration sets.

As the size of LLMs increases, both methods demonstrate improved robustness. However, a notable difference emerges in their performance under varying conditions. While QuIP performs effectively on 2-bit quantization when the evaluation dataset aligns with the calibration set (see Table 2), it experiences a significant performance decline, and may even diverge, when tested on a different dataset. This decline can be attributed to QuIP's dependence on the Hessian matrix derived from

the calibration set, which makes it highly sensitive to changes in the dataset. In contrast, CMPQ exhibits greater resilience, relying only on the average activation $L_2$-norm from the calibration set – a measure that generalizes more robustly across different datasets.

Table 5: Robustness analysis of the calibration dataset. We compare CMPQ with QuIP for 2-bit quantization tasks and report the perplexity differences across two different calibration datasets.

| Models | Eval / Calib | QuIP | | CMPQ | |
|---|---|---|---|---|---|
| | | Wiki | C4 | Wiki | C4 |
| **OPT-2.7B** | Wiki | 32.84 | 242.69 (+204.62) | 29.62 | 27.56 (-0.76) |
| | C4 | >1000 (+1000) | 38.07 | 32.46 (+2.84) | 28.32 |
| **OPT-6.7B** | Wiki | 22.16 | 107.56 (+85.94) | 18.86 | 18.39 (+0.08) |
| | C4 | 22.33 (+0.17) | 21.62 | 18.63 (-0.23) | 18.31 |

### 4.6 ABLATION STUDY OF OUTLIER PROTECTION

In this section, we conduct experiments on the OPT-2.7B and OPT-6.7B models to analyze the impact of two distinct types of outliers on quantization performance. For a consistent comparison, when we remove one type of outlier, we retain the other type, ensuring that it constitutes 0.5% of the entire weight matrix. The results, presented in Table 6, demonstrate that both types of outliers contribute to enhancing the performance of quantized LLMs, particularly in low-bit settings. Notably, protecting both types of outliers yields the best results in general. This is because each type of outlier addresses different aspects: activation-based outliers safeguard salient weights, while quantization-based outliers ensure that the clustering process during quantization is not distorted by extreme values, thereby focusing on the majority of weights. In summary, these two outlier protection strategies complement each other, working in tandem to improve the overall model performance.

Table 6: Ablation of the outlier protection strategies. Best performances are in bold, with underlined text showing the second best.

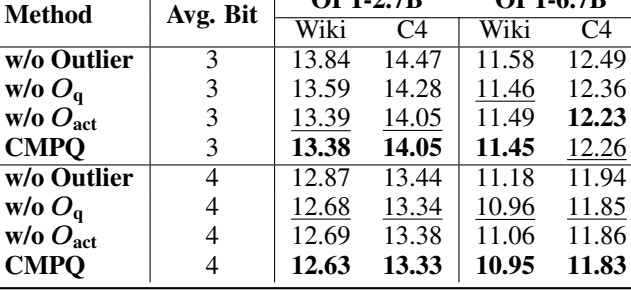

| Method | Avg. Bit | OPT-2.7B | | OPT-6.7B | |
|---|---|---|---|---|---|
| | | Wiki | C4 | Wiki | C4 |
| w/o Outlier | 3 | 13.84 | 14.47 | 11.58 | 12.49 |
| w/o $O_q$ | 3 | 13.59 | 14.28 | 11.46 | 12.36 |
| w/o $O_{act}$ | 3 | 13.39 | 14.05 | 11.49 | **12.23** |
| **CMPQ** | 3 | **13.38** | **14.05** | **11.45** | 12.26 |
| w/o Outlier | 4 | 12.87 | 13.44 | 11.18 | 11.94 |
| w/o $O_q$ | 4 | 12.68 | 13.34 | 10.96 | 11.85 |
| w/o $O_{act}$ | 4 | 12.69 | 13.38 | 11.06 | 11.86 |
| **CMPQ** | 4 | **12.63** | **13.33** | **10.95** | **11.83** |

## 5 CONCLUSION AND FUTURE WORK

In this work, we focused on mixed-precision quantization and aimed to design an algorithm capable of adapting to any bit-width constraint. We observed that different weight channels had varying impacts on model performance, and that activation distributions helped identify salient channels. Building on these insights, we proposed CMPQ, which integrated a channel-wise non-uniform quantization strategy. To further enhance performance, CMPQ introduced two types of outliers that collaboratively preserved critical information. Experimental results showed that CMPQ harnessed the potential of mixed-precision quantization in two key ways: (1) it achieved superior performance in integer-bit quantization tasks, and (2) it delivered significant performance improvements with only a modest increase in memory requirements. In the future, our focus will be on deploying CMPQ-quantized LLMs on real-world devices. This will involve addressing key challenges such as engineering hardware acceleration and designing efficient lookup table kernels to optimize performance.

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

# A APPENDIX

## A.1 IMPACT OF SPARSITY LEVELS OF CMPQ

As discussed in Section 4.6, both activation-based and quantization-based outliers contribute to performance improvements. To evaluate the trade-off between performance and outlier protection ratio, we adjusted the ratio of activation-based outliers from 0.05% to 0.45% and present the perplexity results of the 3-bit quantized OPT-6.7B model on the C4 benchmarks, with varying outlier extraction percentages ranging from 0% to 0.5%, as shown in Fig 4. Notably, while we maintain a fixed protection ratio of 0.5% for quantization-based outliers across all experiments to ensure fair comparisons, the plot reveals that the perplexity gains diminish when the protection ratio exceeds 0.2%. This finding highlights CMPQ's potential to achieve superior performance with reduced storage requirements.

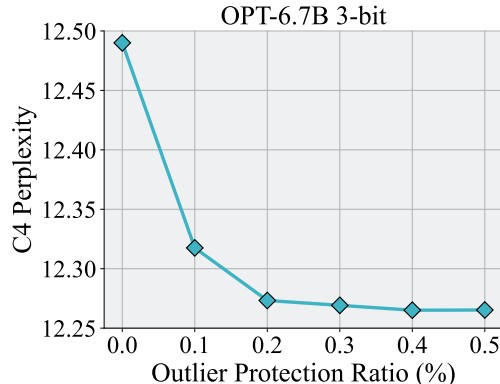

Figure 4: Outlier protection ratio and perplexity trade-off of 3-bit quantized OPT-6.7B model.

## A.2 IMPACT OF NON-UNIFORM QUANTIZATION

In Table 7, we provide a detailed analysis to further clarify the impact of non-uniform quantization. For uniform quantization, we apply the widely used round-to-nearest method with a group size of 128 for channel-wise weight quantization, while preserving 0.5% of activation-based outliers to ensure a fair comparison. Additionally, we report the best perplexity achieved by the baseline methods. As shown in Table 7, across various bit-widths and model sizes, non-uniform quantization consistently outperforms uniform quantization, particularly in extremely low-bit (2-bit) settings. This is because the non-uniform distribution of weights leads to inefficient utilization of the quantization bins in uniform quantization, where some bins may remain underutilized or unused. Interestingly, we also observe that for certain tasks, uniform quantization can improve perplexity (e.g., OPT-13B at 3-bit on WikiText2). In such cases, equipping CMPQ with uniform quantization yields the best performance.

Table 7: Perplexity (↓) comparison on the C4 and WikiText-2 datasets. LLMs are quantized by CMPQ using non-uniform and uniform approaches.

| Method | Avg. Bit | OPT-2.7B | | OPT-6.7B | | OPT-13B | |
| --- | --- | --- | --- | --- | --- | --- | --- |
| | | Wiki | C4 | Wiki | C4 | Wiki | C4 |
| **Baseline** | 2 | >100 | 38.07 | 22.33 | 21.62 | **16.02** | 16.60 |
| **Uniform** | 2 | >100 | 80.52 | >100 | >100 | >100 | >100 |
| **Non-Uniform** | 2 | **32.46** | **28.32** | **18.63** | **18.31** | 16.48 | **16.30** |
| **Baseline** | 3 | 16.32 | 15.63 | 11.41 | 12.30 | 10.50 | 12.39 |
| **Uniform** | 3 | 13.49 | 14.06 | **11.41** | 12.28 | **10.42** | 11.65 |
| **Non-Uniform** | 3 | **13.38** | **14.05** | 11.45 | **12.26** | 10.67 | **11.64** |
| **Baseline** | 4 | 12.69 | 13.48 | 10.93 | 11.86 | 10.21 | 11.28 |
| **Uniform** | 4 | 12.63 | 13.34 | **10.91** | 11.84 | 10.19 | 11.27 |
| **Non-Uniform** | 4 | **12.63** | **13.33** | 10.95 | **11.83** | **10.17** | **11.27** |

## A.3 ADDITIONAL EXPERIMENTAL RESULTS

In Table 8, we present a comparison of quantization results across additional LLMs, including models from the OPT family ranging from 1.3B to 30B parameters, as well as the more recent LLaMA3-7B. Our findings are consistent with the main results, indicating that CMPQ generally outperforms all

baselines. However, we observe that as model size increases and the models become more powerful, the performance gap between different methods narrows. For instance, on the OPT-30B model, CMPQ outperforms baselines in only 2 out of 6 tasks. We attribute this to the differing quantization strategies: while the baselines quantize weights in blocks of 128, CMPQ performs quantization channel-wise. As model size grows, the additional bits allocated by these block-based methods may exceed those introduced by CMPQ, leading to similar performance outcomes. Nevertheless, CMPQ remains competitive in terms of performance while offering lower storage requirements.

Table 8: Additional perplexity (↓) comparison of {2, 3, 4}-bit quantized LLMs on the C4 and WikiText-2 datasets. Bold font indicates the best performance across all methods, while underlined results denote the second-best.

| Method | Avg. Bit | OPT-1.3B | | OPT-13B | | OPT-30B | | LLaMA3-8B | |
|--------|----------|----------|------|---------|------|---------|------|-----------|------|
| | | Wiki | C4 | Wiki | C4 | Wiki | C4 | Wiki | C4 |
| **FP16** | 16 | 14.62 | 14.72 | 10.13 | 11.2 | 9.56 | 10.69 | 6.1 | 9.2 |
| **RTN** | 2 | >1000 | >1000 | >1000 | >1000 | >1000 | >1000 | >1000 | >1000 |
| **GPTQ** | 2 | >1000 | >1000 | 372.68 | 135.48 | 71.7 | 29.59 | 210 | >1000 |
| **AWQ** | 2 | – | – | – | – | – | – | >1000 | >1000 |
| **QuIP** | 2 | 41.64 | **29.78** | **16.02** | 16.6 | **11.48** | 13.55 | **85.1** | 130 |
| **CMPQ** | 2 | **41.58** | 36.67 | 16.48 | **16.3** | 11.53 | **12.89** | 120 | **110** |
| **RTN** | 3 | >1000 | >1000 | >1000 | >1000 | >1000 | >1000 | 27.9 | 110 |
| **GPTQ** | 3 | 21.35 | 21.59 | 11.6 | 13.34 | 10.32 | 12.23 | 8.2 | 13.7 |
| **AWQ** | 3 | 16.32 | 16.28 | 10.67 | 12.61 | 9.85 | **10.96** | 8.2 | 11.6 |
| **QuIP** | 3 | 16.21 | 17.12 | **10.5** | 12.39 | **9.79** | 11.66 | **7.5** | 11.3 |
| **CMPQ** | 3 | **16.07** | **16.08** | 10.67 | **11.64** | 9.83 | 10.98 | 7.89 | **11.3** |
| **RTN** | 4 | 47.62 | 27.2 | 11.32 | 12.35 | 10.77 | 13.52 | 8.5 | 13.4 |
| **GPTQ** | 4 | 15.59 | 16.96 | 10.31 | 12.26 | 9.63 | 11.8 | 7.0 | 10.4 |
| **AWQ** | 4 | 14.94 | 15.04 | 10.22 | 11.28 | **9.59** | 10.75 | 6.6 | 9.4 |
| **QuIP** | 4 | 14.88 | 16.38 | 10.21 | 12.16 | 9.61 | 11.5 | 6.6 | 11.3 |
| **CMPQ** | 4 | **14.84** | **14.99** | **10.17** | **11.27** | 9.61 | **10.74** | **6.5** | **9.39** |

