# OpenReview forum: "Channel-Wise Mixed-Precision Quantization for Large Language Models"
_ICLR.cc/2025/Conference — ICLR 2025 Conference Withdrawn Submission_

### Official Review · Reviewer_yUHF · 2024-11-01

**Soundness:** 3
**Presentation:** 2
**Contribution:** 2
**Rating:** 3
**Confidence:** 4

**Summary:**

The paper presents an approach to quantize large language models (LLMs) with channel-wise mixed-precision quantization. This method allows different channels of neural network layers to be quantized at different precision levels.  Experiments on OPT and LLaMA show that their method outperforms existing quantization techniques.

**Strengths:**

1. The channel-wise mixed-precision quantization can help deal with channel outliers.
2. The experimental results indicate substantial improvements in accuracy after quantization.

**Weaknesses:**

1. There are typography disorders on page 7.
2. The overall process of CMPQ during inference is not well elaborated. For example, how do you perform de-quantization efficiently?
3. The real deployment efficiency is not presented. Since CMPQ sets different precision for each channel, I am worried about its de-quantization overhead during inference.

**Questions:**

See weaknesses above.

---

> ### Author Response · Authors · 2024-11-21
> **Official Comment by Authors**
>
> >**W1.** There are typography disorders on page 7.
> >
> **Response**: We thank the reviewer for this valuable observation. However, after careful review, we could not identify any typography issues on page 7. Could you kindly provide more specific details or examples of the issues you noticed? We would be happy to make the necessary corrections once clarified.
>
> >**W2.** The overall process of CMPQ during inference is not well elaborated. For example, how do you perform de-quantization efficiently?
> >
> **Response**: We thank the reviewer for this valuable suggestion and would like to address the concern as follows: First, the primary focus of CMPQ is leveraging additional storage space to achieve improved model performance, motivating the use of mixed-precision quantization. Our results demonstrate significant performance gains with only a modest increase in memory requirements, surpassing uniform quantization methods. Second, we acknowledge that mixed-precision quantization introduces challenges related to inference efficiency, particularly for de-quantization.
> While a detailed latency analysis of CMPQ is left for future work, we believe that implementation techniques from SqueezeLLM could be adapted to CMPQ. Both methods use channel-wise quantization, and SqueezeLLM leverages lookup tables (LUT) for output channels. Their latency evaluations provide preliminary insights, and we believe this approach can serve as a foundation for extending de-quantization analysis to CMPQ.
>
> >**W3.** The real deployment efficiency is not presented. Since CMPQ sets different precision for each channel, I am worried about its de-quantization overhead during inference.
> >
> **Response**: We thank the reviewer for highlighting this important aspect. As discussed in our earlier response, we propose leveraging insights from SqueezeLLM’s latency analysis to address de-quantization overhead in CMPQ. Specifically, SqueezeLLM demonstrates that a LUT-based non-uniform quantization approach can achieve a 2.4× speedup compared to the FP16 baseline. Additionally, it exhibits comparable latency to the uniform quantization approach GPTQ. Since CMPQ also uses channel-wise quantization, similar lookup-table techniques can be adapted. While a dedicated analysis for CMPQ is not included due to limitations in existing evaluation platforms for mixed-precision methods, the results from SqueezeLLM provide a strong indication of CMPQ’s potential efficiency.

---

> > ### Comment · Reviewer_yUHF · 2024-11-22
> >
> > Thank the authors' response. While I appreciate the authors' efforts, I find that the issue of deployment efficiency remains unaddressed in the current version. Given that the proposed compression method is intended for large language models (LLMs), this is a critical consideration that significantly impacts the practical applicability of the approach. Therefore, I will revise my score to reject.

---

### Official Review · Reviewer_hujJ · 2024-11-02

**Soundness:** 3
**Presentation:** 2
**Contribution:** 1
**Rating:** 3
**Confidence:** 4

**Summary:**

The paper introduces Channel-Wise Mixed-Precision Quantization (CMPQ), a mixed quantization method for large language models (LLMs) that assigns quantization precision in a channel-wise manner based on activation distributions. This approach aims to reduce memory requirements while maintaining performance by adaptively allocating bit precision within each weight channel. CMPQ also employs non-uniform quantization and two outlier extraction methods to preserve critical information, reducing quantization loss. Experiments across various LLMs and datasets indicate that CMPQ outperforms baseline quantization methods, achieving notable memory savings with minimal performance compromise, particularly in tasks with integer-bit and fractional-bit quantization constraints

**Strengths:**

One of the strengths of this paper is its focused and in-depth exploration of outliers in quantization. Unlike many previous methods, which either overlook or handle outliers with basic techniques, CMPQ presents a comprehensive approach by categorizing outliers into activation-based and quantization-sensitive types. This dual approach is a significant step forward, as it ensures that high-impact weights are preserved where they matter most, leading to enhanced performance stability across different quantization levels. By offering a granular solution to outlier management, the paper addresses a critical gap in existing research, showing a well-founded understanding of the practical challenges in mixed-precision quantization.

Another commendable aspect of this paper is its ability to synthesize and build upon various existing quantization techniques, resulting in an integrated solution. The authors leverage insights from both uniform and non-uniform quantization studies, incorporating methods like channel-wise precision allocation and selective outlier retention to maximize CMPQ’s efficacy. This integrative approach not only showcases the strengths of CMPQ but also places it within the broader context of quantization research, illustrating how it enhances or complements other methods. This approach adds robustness to the results, providing a clear, side-by-side comparison with traditional integer-based techniques and showing the benefits of CMPQ’s adaptive precision allocation.

**Weaknesses:**

1. While CMPQ leverages mixed-precision quantization to improve model performance, the paper lacks a rigorous theoretical framework for determining the allocation of different bit widths. Mixed-precision inherently introduces challenges in establishing precise criteria and policies for bit-width allocation across channels, which often leads to overly practical and heuristic-based solutions. Without a solid theoretical foundation, the method risks being overly tailored to specific scenarios, limiting its generalizability and making it challenging to adapt across varying large language models.

2. Mixed-precision quantization introduces unstructured patterns in model weights, which can negatively impact the execution speed on real-world applications, particularly in hardware environments where consistent bit-width allocation is optimal. The paper does not adequately address these potential speed penalties, which are crucial for deployment. Rather than leaving this as a topic for "further work," an initial analysis of the impact on runtime performance would provide a more comprehensive view of CMPQ’s practicality, especially in latency-sensitive applications.

3. The paper primarily demonstrates the effectiveness of CMPQ through Wiki and C4 perplexity scores, which, while relevant, do not provide a comprehensive view of performance for large language models (LLMs) in complex tasks. In current LLM research, deeper benchmarks—such as MMLU, GSM8K, or win-rate metrics against existing models like GPT—are essential to capture a model’s real-world utility and robustness. Over-reliance on perplexity can be misleading, as slight improvements in perplexity may not translate to meaningful gains in more challenging tasks, where small differences in Wiki perplexity can result in significant performance gaps in harder benchmarks. A broader evaluation would provide stronger evidence of CMPQ’s effectiveness in realistic settings and make the findings more comparable with current LLM standards.

**Questions:**

included in Weaknesses.

---

> ### Author Response · Authors · 2024-11-21
> **Official Comment by Authors**
>
> >**W1.** CMPQ lacks a rigorous theoretical framework for allocating bit widths in mixed-precision quantization, leading to heuristic-based solutions that limit generalizability across different models.
> >
> **Response**: We thank the reviewer for raising this insightful concern. However, we would like to make the following clarification. The primary goal of model quantization is to reduce memory consumption during storage while minimizing performance degradation during inference. The majority of the existing methods, including LLM-MQ, rely on empirically driven heuristics, highlighting the current state of research in mixed-precision quantization. Given this, our focus has been on leveraging practical and storage-efficient heuristics to achieve significant performance improvements. CMPQ demonstrates strong performance across LLMs, and this robustness suggests that CMPQ  is adaptable to different architectures and not overly tailored to specific scenarios. While theoretical guarantees are desirable, empirical effectiveness is often more relevant in real-world deployment scenarios, where resource trade-offs are a critical factor.
>
> >**W2.** Mixed-precision quantization introduces unstructured patterns in model weights, which may negatively impact real-world deployment, especially for latency-sensitive applications.
> >
> **Response**: We thank the reviewer for highlighting this important aspect. We would like to address your concern as follows: First, the primary objective of CMPQ is to make full use of the moderate storage space to achieve significant performance improvements over integer quantization. For example, in scenarios where hardware supports an average precision of 2.2 bits, CMPQ can efficiently utilize the extra 0.2 bits per parameter to enhance performance—something beyond the scope of any integer bit-width allocation methods. Second, we acknowledge that deploying mixed-precision models on hardware introduces complexities. A detailed analysis of latency for CMPQ would require specific platform support, which is currently lacking for mixed-precision methods in tools like GPTQ. Third, despite the lack of direct support, the implementation of SqueezeLLM offers a relevant baseline since CMPQ and SqueezeLLM both quantize LLMs in a channel-wise way. SqueezeLLM uses channel-wise quantization with lookup tables for output channels, and their latency evaluations provide preliminary insights for extending such analyses to CMPQ. We will discuss this connection in the revised manuscript and outline directions for future latency evaluations
>
> >**W3.**  Evaluation is limited to Wiki and C4 perplexity scores, missing broader benchmarks like MMLU or GSM8K that are crucial for assessing LLM robustness in complex tasks. Broader benchmarks would strengthen the evidence of CMPQ's effectiveness.
> >
> **Response**: We thank the reviewer for this valuable suggestion. In response, we have conducted additional experiments of LLaMA2-7B on the MMLU benchmark, which evaluates a broader range of real-world capabilities. These results, summarized in the table below, demonstrate that CMPQ continues to achieve superior performance, further validating its robustness and effectiveness.
> effectiveness and robustness. We will include these findings in the revised manuscript to provide a more comprehensive evaluation of CMPQ.
> Bit  | Category           | FP16  | RTN   | GPTQ  | QUIP  | AWQ   | CMPQ  |
> |------|--------------------|-------|-------|-------|-------|-------|-------|
> | 3    | STEM               | 37.31 | 27.93 | 28.24 | 27.27 | 31.74 | 32.54 |
> | 3    | Social Sciences    | 52.91 | 23.63 | 25.67 | 30.81 | 36.59 | 43.26 |
> | 3    | Humanities         | 43.04 | 24.87 | 31.69 | 27.08 | 30.86 | 36.34 |
> | 3    | Other              | 53.82 | 24.12 | 32.92 | 25.51 | 38.09 | 45.99 |
> | 3    | Average            | 46.46 | 25.08 | 29.85 | 27.57 | 33.98 |**39.27**|
>
> Bit  | Category           | FP16  | RTN   | GPTQ  | QUIP  | AWQ   | CMPQ  |
> |------|--------------------|-------|-------|-------|-------|-------|-------|
> | 4    | STEM               | 37.31 | 34.59 | 34.13 | 36.25 | 36.24 | 36.48 |
> | 4    | Social Sciences    | 52.91 | 46.05 | 43.87 | 48.07 | 50.31 | 50.57 |
> | 4    | Humanities         | 43.04 | 39.62 | 38.13 | 39.83 | 41.72 | 41.91 |
> | 4    | Other              | 53.82 | 47.78 | 45.59 | 49.07 | 52.22 | 52.16 |
> | 4    | Average            | 46.46 | 41.83 | 40.25 | 43    | 44.85 | **45**   |

---

> > ### Comment · Reviewer_hujJ · 2024-11-22
> >
> > Thank you for your thoughtful comments. Regarding W1, while I largely agree with the concern, it does not substantially alter my perspective on the novelty of the paper. The addition of W3 results is a positive step, but it is somewhat expected that a more unstructured compression method would outperform others in benchmarks. This makes addressing W2 even more critical, as demonstrating that the approach does not introduce significant overhead is essential for practical relevance. Therefore, my overall evaluation of the paper remains consistent.

---

### Official Review · Reviewer_u3Qr · 2024-11-03

**Soundness:** 1
**Presentation:** 2
**Contribution:** 1
**Rating:** 3
**Confidence:** 4

**Summary:**

In this manuscript, the authors introduce a channel-wise mixed-precision quantization (CMPQ) method designed to efficiently compress large language models. Unlike previous mixed-precision quantization approaches, which use a fixed bit allocation per layer, the proposed method enables different bit allocations for individual channels, inspired by insights from the 'Massive Activation' paper. By incorporating several outlier protection techniques, the authors demonstrate that a modest increase in memory usage can lead to significant improvements in accuracy across various language benchmarks.

**Strengths:**

The motivation is compelling, as Figure 2 clearly illustrates that different channels, particularly those associated with tokens exhibiting massive activation, have varying impacts on overall accuracy. This suggests that using uniform bit allocations across all channels could be highly inefficient. To address this, the authors incorporate established outlier detection techniques to manage these outlier variations effectively. Their results show that only a few channels require different bit allocations, while most channels remain relatively insensitive to changes introduced by these outlier handling methods.

**Weaknesses:**

Unfortunately, this manuscript faces several critical issues that limit its practicality. Here are some examples to illustrate these concerns:

- The manuscript does not address how to accelerate the proposed quantization method effectively. Since the bit allocations vary across different channels, parallel operations that process multiple channels simultaneously could introduce significant performance bottlenecks, requiring special handling to mitigate these issues.

- The selection of outlier thresholds (such as preserving 0.45% of the largest values in FP16 precision) appears overly empirical. These thresholds are either adopted from prior work without sufficient context or presented without robust justification, undermining the method's theoretical foundation.

- The approach relies on augmenting quantized weights with a small amount of FP16 data, but the manuscript fails to discuss the impact of this additional FP16 memory on overall computational complexity or performance. The experimental results do not account for the implications of handling this extra FP16 data.

- The manuscript uses only perplexity (PPL) as the metric for evaluating quantization quality. However, more comprehensive metrics, such as MMLU, are necessary to better reflect the capabilities of modern large language models, as PPL alone does not provide a complete assessment of performance.

**Questions:**

What is the performance of LLMs when using the proposed quantization method? How much degradation occurs as a result of the additional FP16 computations?

---

> ### Author Response · Authors · 2024-11-21
> **Official Comment by Authors**
>
> >**W1.** The manuscript does not address how to accelerate the proposed quantization method effectively. Since the bit allocations vary across different channels, parallel operations that process multiple channels simultaneously could introduce significant performance bottlenecks, requiring special handling to mitigate these issues.
> >
> **Response**: We thank the reviewer for highlighting this important aspect. First, we emphasize that the primary focus of our work is on leveraging additional storage space for improved model performance, which motivates the adoption of mixed-precision quantization. Our results demonstrate significant performance gains with only a modest increase in memory requirements, which is beyond the reach of any uniform quantization methods. Second, we acknowledge that mixed-precision quantization inherently introduces challenges for integrating parallel operations, but this primarily affects inference latency rather than model performance. Our proposed method is most effective when improved model performance is the ultimate goal, while some processing latency is allowed. We are not claiming CMPQ improves both inference performance and latency over the existing methods. In our study, all comparisons with baselines are conducted under the same conditions to ensure fairness. We agree that addressing the acceleration of CMPQ is a valuable avenue for future research. However, existing platforms like GPTQ currently lack support for evaluating mixed-precision methods, posing limitations for comprehensive latency analysis. Nonetheless, we believe that the implementation techniques used in SqueezeLLM could be adapted for CMPQ with some modifications since CMPQ and SqueezeLLM both quantize LLMs in a channel-wise way. SqueezeLLM uses channel-wise quantization with lookup tables for output channels, and their latency evaluations provide preliminary insights for extending such analyses to CMPQ.
>
> >**W2.** The selection of outlier thresholds (such as preserving 0.45% of the largest values in FP16 precision) appears overly empirical. These thresholds are either adopted from prior work without sufficient context or presented without robust justification, undermining the method's theoretical foundation.
> >
> **Response**: We thank the reviewer for this observation. We would like to clarify that our approach to selecting outlier thresholds aligns with existing works such as SqueezeLLM, which similarly protects 0.45% of the largest value outliers and 0.05% sensitivity-based outliers. For fair comparisons with these baselines, we also protect a total of 0.5% outliers, as described in the paper. Furthermore, AWQ highlights that salient channels based on activations are typically under 1%, which positions our threshold of 0.45% well within this range. Given the complexity of large language models, deriving a precise theoretical justification for the proportion of outliers to protect is challenging. Consequently, most existing methods, including ours, are based on empirical observations. The superior performance of CMPQ across diverse models and tasks demonstrates the robustness of our 0.45% + 0.05% threshold selection. While empirical, our approach has proven effective and adaptable to various scenarios.
>
> >**W3.** The approach relies on augmenting quantized weights with a small amount of FP16 data, but the manuscript fails to discuss the impact of this additional FP16 memory on overall computational complexity or performance. The experimental results do not account for the implications of handling this extra FP16 data.
> >
> **Response**: We thank the reviewer for pointing this out. First, it is important to note that many existing methods, including SqueezeLLM and LLM-MQ, also protect 0.5% or more outliers, resulting in comparable computational complexity. Thus, the impact of additional FP16 memory on overall computational complexity in CMPQ aligns with these baselines. More specifically, as discussed in SqueezeLLM, keeping a small amount of FP16 data only adds around 10% latency overhead in lookup-table-based dequantization. Second, as detailed in Appendix A.1, we evaluate the impact of FP16 outlier ratios on performance and identify 0.5% protection as optimal. This ensures a fair comparison with baselines. Regarding storage requirements, CMPQ introduces an overhead of (16–3) × 0.5% = 0.065 bits for 3-bit quantization. In contrast, methods like AWQ or GPTQ with a group size of 128 require an average of 3.24 bits, as reported in SqueezeLLM. Therefore, CMPQ introduces comparable or even lower memory overhead than existing baselines.

---

> > ### Comment · Reviewer_u3Qr · 2024-11-24
> > **Reponse to the authors**
> >
> > I appreciate the authors' efforts to address my concerns. However, several practical yet critical issues remain unresolved.
> >
> > In today's highly parallel computing systems, even small overheads can significantly reduce overall throughput. Consequently, the accuracy or task performance of the proposed models should be compared to that of larger models that are quantized without introducing additional overhead.
> >
> > Without detailed information regarding the overhead introduced by the proposed approach, particularly in terms of latency and throughput, it is challenging to fully evaluate this work. Any overhead that disrupts overall parallelism must be addressed with utmost care.
> >
> > Therefore, I maintain my original score.

---

> ### Author Response · Authors · 2024-11-21
> **Official Comment by Authors**
>
> >**W4.** The manuscript uses only perplexity (PPL) as the metric for evaluating quantization quality. However, more comprehensive metrics, such as MMLU, are necessary to better reflect the capabilities of modern large language models, as PPL alone does not provide a complete assessment of performance.
> >
> **Response**: We thank the reviewer for this valuable suggestion. We have conducted additional experiments of LLaMA2-7B on MMLU tasks, and the results are summarized in the table below. These results show that CMPQ continues to exhibit superior performance on MMLU metrics, further validating its effectiveness and robustness. We will include these findings in the revised manuscript to provide a more comprehensive evaluation of CMPQ.
> effectiveness and robustness. We will include these findings in the revised manuscript to provide a more comprehensive evaluation of CMPQ.
> Bit  | Category           | FP16  | RTN   | GPTQ  | QUIP  | AWQ   | CMPQ  |
> |------|--------------------|-------|-------|-------|-------|-------|-------|
> | 3    | STEM               | 37.31 | 27.93 | 28.24 | 27.27 | 31.74 | 32.54 |
> | 3    | Social Sciences    | 52.91 | 23.63 | 25.67 | 30.81 | 36.59 | 43.26 |
> | 3    | Humanities         | 43.04 | 24.87 | 31.69 | 27.08 | 30.86 | 36.34 |
> | 3    | Other              | 53.82 | 24.12 | 32.92 | 25.51 | 38.09 | 45.99 |
> | 3    | Average            | 46.46 | 25.08 | 29.85  | 27.57 | 33.98 | **39.27** |
>
> Bit  | Category           | FP16  | RTN   | GPTQ  | QUIP  | AWQ   | CMPQ  |
> |------|--------------------|-------|-------|-------|-------|-------|-------|
> | 4    | STEM               | 37.31 | 34.59 | 34.13 | 36.25 | 36.24 | 36.48 |
> | 4    | Social Sciences    | 52.91 | 46.05 | 43.87 | 48.07 | 50.31 | 50.57 |
> | 4    | Humanities         | 43.04 | 39.62 | 38.13 | 39.83 | 41.72 | 41.91 |
> | 4    | Other              | 53.82 | 47.78 | 45.59 | 49.07 | 52.22 | 52.16 |
> | 4    | Average            | 46.46 | 41.83 | 40.25 | 43    | 44.85 | **45**   |
>
> >**Q1.** What is the performance of LLMs when using the proposed quantization method? How much degradation occurs as a result of the additional FP16 computations?
> >
> **Response**: We thank the reviewer for this question. The performance of LLMs using CMPQ is thoroughly reported in Section 4.2 and Appendix A.3 of the manuscript. The impact of additional FP16 computations is discussed in Section 4.6 and Appendix A.1. In summary, CMPQ demonstrates significant performance improvements with minimal degradation, even with the inclusion of additional FP16 computations.

---

### Official Review · Reviewer_dWAS · 2024-11-03

**Soundness:** 2
**Presentation:** 3
**Contribution:** 2
**Rating:** 5
**Confidence:** 4

**Summary:**

This work proposes a Channel-Wise Mixed-Precision Quantization method to optimize quantization by leveraging activation distributions, offering a more fine-grained approach compared to traditional layer-wise quantization.

**Strengths:**

The proposed method shows better performance over GPTQ/AWQ, etc., on various large language models.

**Weaknesses:**

1.  The contribution of this paper is relatively minor and could be considered essentially an extended version of SmoothQuant to a limited extent.

2.  By comparing Tables 1 and 2, the performance improvements over the baseline are not stable and, in some cases, are worse. For example, under 4-bit precision on the C4 dataset for LLaMA2-7B, compared with QuIP (7.10 vs. 6.69), more discussion is required.

3.  The ablation results show that w/o Oact has little effect, indicating that output activation is not crucial for performance.

**Questions:**

Please refer to the weaknesses.

---

> ### Author Response · Authors · 2024-11-21
> **Official Comment by Authors**
>
> >**W1.** The contribution of this paper is relatively minor and could be considered essentially an extended version of SmoothQuant to a limited extent.
> >
> **Response**: We thank the reviewer for their feedback and for raising this point. However, we would like to clarify that **our work is not an extension of SmoothQuant**. First, as acknowledged by other reviewers (e.g., Reviewer hujJ), our research addresses a critical gap in existing quantization techniques, offering an integrated solution that builds upon and unifies various approaches. Second, unlike SmoothQuant, which quantizes activations and weights simultaneously, CMPQ adopts a distinct research direction called weight-only quantization (introduced in line 110). Specifically, CMPQ employs non-uniform, channel-wise weight quantization with varying precision, as opposed to SmoothQuant’s uniform quantization approach (discussed in Section 3.1.2). Lastly, CMPQ is tailored for scenarios where model performance is highly sensitive to quantization. Our method achieves significant performance improvements with only a modest increase in memory requirements (non-integer bit quantization), a capability beyond the reach of integer-bit methods like SmoothQuant. We believe these distinctions highlight CMPQ’s novelty and contribution, with unequivocal distinctions to SmoothQuant.
>
> >**W2.** By comparing Tables 1 and 2, the performance improvements over the baseline are not stable and, in some cases, are worse. For example, under 4-bit precision on the C4 dataset for LLaMA2-7B, compared with QuIP (7.10 vs. 6.69), more discussion is required.
> >
> **Response**: We thank the reviewer for pointing this out. We apologize that there was an error in reporting the 4-bit precision result for the C4 dataset with LLaMA2-7B. Specifically, the value of 6.69 for QuIP is counterintuitive, as it surpasses the FP16 precision result of 6.97 (Table 2). After rechecking, the correct CMPQ result for this setting is 7.12. With this correction, **CMPQ outperforms QuIP in this task**. We will update Table 2 and the corresponding Figure 3 in the revised manuscript. Additionally, while CMPQ does not outperform baselines in 2 out of 24 tasks, it consistently achieves superior results across 22 tasks, reflecting its robustness. Moreover, the core contribution of CMPQ lies in its ability to fully leverage additional storage space to deliver significant performance improvements. For example, in Figure 3, with LLaMA2-7B, CMPQ achieves a 30% improvement in perplexity (15.97 → 11.11) with only a 10% increase in storage overhead at 2-bit precision. Such enhancements are beyond the scope of integer-bit quantization methods like QuIP. We will emphasize these aspects in the revised version for better clarity.
>
> >**W3.** The ablation results show that w/o Oact has little effect, indicating that output activation is not crucial for performance.
> >
> **Response**: We thank the reviewer for this observation. However, we would like to clarify that in Figure 6, when removing one type of outlier, we increase the protected outlier of another type to 0.5%. As shown in our results, in the 3-bit quantization task, protecting 0.5% of $O_{q}$ yields better performance than protecting 0.5% of $O_{act}$ on specific tasks. However, this trend reverses for the 4-bit quantization results, where protecting $O_{act}$ shows greater importance. The combined effectiveness of CMPQ, which integrates both types of outliers, underscores that both $O_{act}$ and $O_{q}$ are critical for performance in different scenarios. We will clarify this interplay further in the revised manuscript.

---

> > ### Comment · Reviewer_dWAS · 2024-11-27
> >
> > Thanks. I appreciate the authors' effort. I decided to retain my score.

---

### Note · Authors · 2025-01-21

I have read and agree with the venue's withdrawal policy on behalf of myself and my co-authors.